# Minimizing Global Buffer Access in a Deep Learning Accelerator Using a Local Register File with a Rearranged Computational Sequence

**DOI:** 10.3390/s22083095

**Published:** 2022-04-18

**Authors:** Minjae Lee, Zhongfeng Zhang, Seungwon Choi, Jungwook Choi

**Affiliations:** Department of Electronic Engineering, Hanyang University, Seoul 04763, Korea; minjae.lee@dsplab.hanyang.ac.kr (M.L.); zhongfeng.zhang@dsplab.hanyang.ac.kr (Z.Z.)

**Keywords:** deep learning accelerator, field-programmable gate array (FPGA), local register file, rearrangement of computational sequence

## Abstract

We propose a method for minimizing global buffer access within a deep learning accelerator for convolution operations by maximizing the data reuse through a local register file, thereby substituting the local register file access for the power-hungry global buffer access. To fully exploit the merits of data reuse, this study proposes a rearrangement of the computational sequence in a deep learning accelerator. Once input data are read from the global buffer, repeatedly reading the same data is performed only through the local register file, saving significant power consumption. Furthermore, different from prior works that equip local register files in each computation unit, the proposed method enables sharing a local register file along the column of the 2D computation array, saving resources and controlling overhead. The proposed accelerator is implemented on an off-the-shelf field-programmable gate array to verify the functionality and resource utilization. Then, the performance improvement of the proposed method is demonstrated relative to popular deep learning accelerators. Our evaluation indicates that the proposed deep learning accelerator reduces the number of global-buffer accesses to nearly 86.8%, consequently saving up to 72.3% of the power consumption for the input data memory access with a minor increase in resource usage compared to a conventional deep learning accelerator.

## 1. Introduction

Recently, many studies have been published regarding deep neural networks (DNNs) for application in various areas, such as in classification [1], object detection [2], speech recognition [3], and image/video recognition [4,5]. This study addresses the problem of executing the required tasks of a DNN with a dedicated hardware accelerator. Although a central processing unit (CPU) or digital signal processor (DSP) can be considered for some applications instead of the accelerator [6,7], CPUs and DSPs are not sufficiently fast to satisfy the timing requirements for the real-time processing of heavy-duty tasks owing to their high latency [8]. In addition, a graphic processing unit consisting of a large number of powerful processors can be an alternative only for limited cases, e.g., when power consumption is not a critical issue [9,10,11].

Recalling that the main functionality of a DNN is to perform convolution between input data and weight(s), we present a novel procedure for executing convolutional operations using a dedicated hardware accelerator implemented on a general-purpose field-programmable gate array (FPGA). Particularly, this paper presents a method for minimizing the number of global buffer accesses within the accelerator by utilizing a local register file shared across the multiply-accumulate (MAC) units of a 2D computation array.

Notably, most deep learning accelerators consume far more power on data movements than on arithmetic operations [12,13,14]. In [15,16], the authors presented data reuse through loop unrolling, interchanging, and tiling, whereas [17,18,19,20,21,22] introduced a more specific method of data reuse, namely, data reuse via row-, output-, or weight-stationary dataflows. Furthermore, ref. [22] disclosed all the procedures for data reuse based on the weight-stationary method. However, in all these studies, a significant number of non-stationary input data pixels are not reused at a MAC unit level. Therefore, input data should be reread from the global buffer, even though they were already read for the previous computations.

Noting that a local register file consumes much less power than a global buffer, the total power consumption of the data movements can be significantly reduced by substituting the proposed local register file access for the global buffer access. However, the resource overhead of register files is greater than the global buffer overhead. Thus, it is costly to maintain a sizeable local register file within each MAC unit. For example, ref. [17] could only equip a small number of registers in each MAC to enable limited input data reuse.

We propose a rearranged computation sequence that allows sharing a single register file across multiple MAC units to alleviate this limitation. Specifically, the convolution computation along the output channel dimension is spatially mapped across the MAC column. A local register file shared by a column of MAC units keeps a block of input data pixels temporarily reused over pixel-wise convolution computation. Since all the MAC columns operate on the same pixel (across the different input channels), a single input index unit (called Input Index Controller(IIC)) controls all the local register files, reducing the control overhead of the proposed mechanism. Our FPGA implementation and the performance evaluation on popular DNNs validate that the proposed method incurs only about 10% of hardware resources while reducing the global buffer access by 45% to 86%, achieving up to 72% of power savings in data movements.

The remainder of this paper is organized as follows. In Section 2, we present an extensive analysis of the spatial and temporal data reuse achievable with the two-dimensional multiplier-accumulator (2-D MAC) array; based on this, we summarize the barriers that need to be overcome to fully exploit the merits of data r euse. In Section 3, we present the proposed technique for minimizing global buffer access by using a local register file based on a novel rearrangement of the computational sequence required in a given DNN. In Section 4, the proposed accelerator is implemented using a commercial FPGA to verify the functionality and resource utilization of the convolutional operations employing the proposed rearrangement. In Section 5, we present simulation results to demonstrate the superiority of the proposed accelerator in comparison to conventional techniques [17,18,19,20,21,22]. Finally, Section 6 concludes the paper.

## 2. Global Buffer Access within a Deep Learning Accelerator

To minimize the global buffer access required for convolutional operations, it is necessary to analyze the global buffer access pattern in detail. After summarizing the spatial and temporal data reuse in Section 2.1 and Section 2.2, respectively, which are partially presented in the state-of-the-art techniques [17,18,19,20,21,22], we present the barriers in the conventional methods that should be overcome to practically exploit both spatial and temporal data reuse in Section 2.3, leading to the minimization of global buffer access as described in Section 3.

### 2.1. Spatial Data Reuse Using Two-Dimensional Multiplier-Accumulator (2-D MAC) Array Structure

This subsection presents how data reuse can be achieved by using parallel processing with the 2-D MAC array structure. The data reuse obtained with the 2-D MAC array can be considered a spatial gain because the data reuse capability is determined by the 2-D MAC array size [15,16].

To analyze the spatial data reuse using a 2-D MAC array structure, we take a simple example, as shown in Figure 1 [22]. Figure 1 illustrates a 2-D MAC array consisting of 16×128 MAC units, each of which performs a convolution between each input data and weight. For simplicity, but without loss of generality, it can be assumed that the numbers of channels for both the input data and each weight are set to 128, with the number of weights being 16.

For the convolutional operation at each MAC to occur correctly, the memory control shown in Figure 1 is provided in such a way that both the input data and weight are fetched from the global buffer and transferred into the MAC array in accordance with the preset operational order. In other words, the key operation of the memory control is to generate a data sequence in accordance with the preset operational order; this is accomplished by generating the address of the global buffer correctly. Once the input data and weight are transferred from the global buffer into the MAC array, each MAC operator performs multiplication, and the final output feature is generated by accumulating the multiplication results.

The MAC unit in the *i*th row of the *j*th column of the 2-D array provides the convolution between the input data and *i*th weight, where both the input data and weight correspond to the *j*th channel. The operation at each of the 16×128 MACs occurs for each pixel of the input data during the corresponding convolutional window.

Because each column of the 2-D MAC array represents the corresponding channel for both the input data and weights, the *j*th channel input data are repeatedly used for the convolution with each of the 16 weights in the *j*th column, whereas j runs from 1 to 128, as in the example shown in Figure 1. In other words, the input data can be reused for each of the 16 weights in each column. In particular, data reuse can be accomplished for each of the 128-channel input data at each column for as many rows in a given 2-D MAC structure: 16 in the example of Figure 1.

As discussed above, the data reuse provided by the 2-D MAC array is available for each column of the array, meaning that the data reuse effect is equivalent only to the case of the 1-D MAC array with the same number of rows.

To expand the data reuse, the procedure of computing the convolution at each MAC should be modified in such a way that data reuse can be provided for each row, as well as for each column. To allow data reuse for a given weight along each row, the weight should remain the same in each row. The weight value at each row is taken from the corresponding channel because the weight should be convolved with the input data of the same channel. For the weight to remain the same in each row, each of the 128 columns must represent a single channel. To accomplish this, instead of performing row-wise parallel processing with each of the 128-channel data, parallel processing should be performed with the different pixel input data of the same channel at each row of the 2-D MAC array. Then, the data reuse for the input data (as well as that for the weight) can be accomplished for both the columns and rows of the 2-D MAC array.

To involve the input data corresponding to all of the different pixels in each row, however, is to increase the width of the output feature pixels as well. This would result in a considerable increase in the buffer size required for storing the partial sums corresponding to each of the output feature pixels [15], potentially imposing a serious limit on hardware implementation [23,24]. Unless this problem is resolved, the data reuse factor cannot be set to a sufficiently high value in all conventional methods [17,18,19,20,21,22]. Furthermore, as mentioned earlier in this subsection, the spatial data reuse factor must be fixed depending on the size of the 2-D MAC array. As will be shown later in Section 3, we present a new technology to allow the data reuse factor to be arbitrarily set by introducing a local register file whose size can be freely set.

### 2.2. Temporal Data Reuse Using Controllable Output Feature Pixels

In contrast to the spatial data reuse discussed in Section 2.1, this subsection presents how to reuse a given dataset at different moments such that data reuse can be temporally achieved. This allows for data reuse regardless of the structure and/or size of the 2-D array [22].

To achieve temporal data reuse, after finishing the convolution of each pixel of the input data with a given pixel of weight, the next pixel of the input data should be processed with the present pixel of the weight instead of applying the next pixel weight onto the next input data pixel. This means that a preset number of input data pixels will be applied sequentially for each weight pixel. By doing so, the same weight pixel does not have to be read again from the global buffer for processing with the next input data pixel. In other words, each weight pixel is processed with all of the input data pixels in advance. The number of input data pixels to be convolved with each weight pixel will later be denoted as the “temporal data reuse factor”. Notably, the temporal data reuse factor can be set to an arbitrary value; it is set to 16 for the procedures described in Section 3 and in our hardware implementation introduced in Section 4.

The spatially obtained columnwise data reuse shown in Section 2.1 is predetermined by the number of rows of the given 2-D MAC array structure. In contrast, the temporal data reuse factor discussed above can be determined arbitrarily, regardless of the 2-D MAC array structure. In particular, the width of the output feature pixels, which is determined by the number of input data pixels to be processed with each weight pixel, can be set arbitrarily as desired. This value is the temporal data reuse factor. Although it is set to 16 in both the procedures described in Section 3 and in the example of our hardware implementation shown in Section 4, it can be set arbitrarily, as desired.

### 2.3. Global Buffer Access Pattern

As summarized in Section 2.1 and Section 2.2, spatial data reuse can be obtained in accordance with the given 2-D MAC array structure, whereas temporal data reuse is applicable regardless of the MAC array structure. Because the two different types of data reuse methods are independent of each other, they can be implemented together on a given 2-D MAC array. More specifically, it was demonstrated in [22] that both spatial data reuse and temporal data reuse can be exploited for the input data and weights, respectively, when implementing a convolutional accelerator with a 2-D MAC array. As mentioned earlier, however, the width of the output feature pixels increases as either the spatial or temporal data reuse factor is increased.

In this subsection, we analyze the global buffer read pattern when both the spatial and temporal data reuses are simultaneously exploited. The objective is to find the input data pixels commonly used for convolutional operations with different weight pixels. Using the analysis given in this subsection, we suggest a novel method of exploiting both spatial and temporal data reuse, in which the latter allows for the reuse of both the input data and weight values. Notably, temporal data reuse was allowed only for the weight pixels in previous works [17,18,19,20,21,22].

In principle, each convolutional operation consists of two steps: first, to multiply the input data pixels by the weight pixels correspondingly and then, to sum up the multiplication results. As a result of this operation, a corresponding output feature pixel is generated. This operation should be repeated for the entire set of input data pixels. However, to apply the method of temporal data reuse, a number of input data pixels are first multiplied by a given weight pixel. This operation is repeated for every weight pixel. The output feature pixels in this case cannot be obtained until the multiplication between the weight pixel and each of the input data pixels is completed. The number of input data pixels processed with each weight pixel, i.e., the temporal data reuse factor, is predetermined, as discussed in Section 2.2 and set to 16 in our implementation, as discussed in Section 3 and Section 4.

It can be assumed that each of the 16 weights shown in Figure 1 consists of nine (= 3×3) pixels, as shown on the right-hand side of Figure 2. To exploit the temporal data reuse with a width of 16, 16 input data pixels should be read from the global buffer to be convolved with the corresponding weight pixel. Figure 2 shows how the input data should be read to provide temporal data reuse with a reuse factor of 16. It can be observed that each set of the 16 input data pixels, {(0,0), (0,1), …, (0,15)}, {(0,1), (0,2), …, (0,16)}, and {(0,2), (0,3), …, (0,17)}, is convolved with the corresponding weight pixels, (0,0), (0,1), and (0,2), during the periods of $t_{0}$, $t_{1}$, and $t_{2}$, respectively. In other words, to read each set of 16 input data pixels, {(0,0), (0,1), …, (0,15)}, {(0,1), (0,2), …, (0,16)}, and {(0,2), (0,3), …, (0,17)}, from the global buffer during $t_{0}$, $t_{1}$, and $t_{2}$, respectively, IIC generates the addresses of the corresponding data.

In fact, Figure 2 explicitly shows which pixels of the 16 input data pixels out of the block of 3×18 pixels are to be read from the global buffer to be convolved with which one of the weight pixels during that time period. The operational procedure shown in Figure 2 represents the convolution between each of the 16 input data pixels and a corresponding weight pixel for producing 16 output feature pixels. In other words, the operational procedure shown in Figure 2 corresponds only to the convolution for a single block of 3×18 input data pixels to produce 16 output feature pixels. Here, we define the term input data block to denote the number of input data pixels needed to generate 16 output feature pixels. In general, for the convolutional operations shown in Figure 2, one input data block includes $N_{wy}\times \left(P+\left(N_{wx}-1\right)\right)$ pixels, where P denotes the temporal data reuse factor with the weight dimension being $N_{wy}\times N_{wx}$.

Let us consider an arbitrary size of input data, for example, the case of 6×21 input data pixels, as shown in Figure 3. The objective is to find the input data read patterns among the different blocks. Out of the 6×21 input data pixels shown in Figure 3, we take three example blocks of the input data pixels. The first block shown at the top of Figure 3 corresponds to the block of 3×18 input data pixels consisting of three rows of {(0,0), (0,1), …, (0,17)}, {(1,0), (1,1), …, (1,17)}, and {(2,0), (2,1), …, (2,17)}. The second block shown at the center of Figure 3 corresponds to a block of 3×20 input data pixels, consisting of three rows of {(0,16), (0,17), …, (0,20), (1,0), (1,1), …, (1,14)}, {(1,16), (1,17), …, (1,20), (2,0), (2,1), …, (2,14)}, and {(2,16), (2,17), …, (2,20), (3,0), (3,1), …, (3,14)}. The third block shown at the bottom of Figure 3 corresponds to a block of 3×20 input data pixels, consisting of three rows of {(1,13), (1,14), …, (1,20), (2,0), (2,1), …, (2,11)}, {(2,13), (2,14), …, (2,20), (3,0), (3,1), …, (3,11)}, and {(3,13), (3,14), …, (3,20), (4,0), (4,1), …, (4,11)}.

Now, let us take a closer look at the convolutional operations corresponding to each of the three blocks of input data pixels. The input data pixels shown on the right-hand side of Figure 3 are convolved with the first-row weight pixels. Our explanation here is given only for the convolution with the first-row weight pixels because the convolutional operations corresponding to the second- and third-row weight pixels are exactly the same as those for the first-row weight pixels.

First, when the block is given in an entire rectangular shape, as in the case of the first block of 3×18 pixels shown at the top of Figure 3, the input data read pattern is determined in such a way that 15 out of 16 input data pixels are used in common for the convolution with two consecutive weight pixels if the two weight pixels are in the same row. In particular, after the 16 input data pixels, for example, {(0,0), (0,1), …, (0,15)}, have been read from the global buffer and processed for the convolution with the weight pixel of (0,0), the input data pixels of {(0,1), (0,2), …, (0,15), (0,16)} should be read from the global buffer for the convolution with the weight pixel of (0,1). Consequently, 15 input data pixels out of 16 are repeatedly read from the global buffer. This global buffer read pattern is repeated for every weight pixel in each row.

In contrast, for the convolution of the second or third block of input data pixels, as shown at the center and bottom of Figure 3, respectively, the input data read pattern is quite different. Figure 3 shows which input data pixels are used in common for the convolution with two consecutive weight pixels in the three different cases. Although the global buffer read patterns vary significantly between the three cases, there are many input data pixels used in common for the convolution with two consecutive weight pixels. In the following section, we present how to minimize global buffer access by exploiting the analysis of the global buffer read pattern.

Figure 4 shows the entire convolutional operation comprising all of the input data blocks. It is assumed that the dimensions of the entire input dataset have been given in such a way that each of the input data blocks is solely determined in the form of an entire rectangular shape, as in the first block shown at the top of Figure 3. This condition can be satisfied if the width of the input data pixels is $16i+N_{wx}$ for i = 0, 1, 2, …when the temporal data reuse factor is 16, with the dimensions of the weight pixels being $N_{wy}\times N_{wx}$. In general, the condition for each input data block to be the type of the first block shown at the top of Figure 3 can be satisfied if the width of the output feature pixels is set as a multiple of the temporal data reuse factor. Table 1 summarizes all the indices used in the convolutional operations discussed herein.

Thus far, we have observed how each of the input data pixels in a given block should be read from the global buffer for convolution with the corresponding weight pixel. From this observation, it has been found that some of the input data pixels are used repeatedly for the convolution with the next weight pixel, meaning that these overlapping input data pixels can be reused such that they do not have to be read again from the global buffer. However, it has also been found that the pattern of the repeated input data pixels varies depending on which input data block is to be convolved with the present weight pixel.

To add to the analysis of the global buffer read pattern within a given block of input data pixels as discussed herein, the global buffer read pattern among the interblock operations can be observed. In other words, we want to find which input data pixels out of those read for the convolutional operations for one block of input data pixels can be reused for the convolutional operations for the next block of input data pixels. In Figure 3, the input data pixels inside the yellow box, for example, are used repeatedly for both the first and second blocks. Similarly, the input data pixels inside the blue box are used for both the second and third blocks of input data pixels. In other words, the six input data pixels located at the last two columns of every input data block are always identical to those located at the first two columns of the next block, thereby indicating that they are read repeatedly for the convolutional operations for the two blocks.

In addition to the six input data pixels, it can be observed from Figure 3 that quite a few other groups of input data pixels are used in common for the convolutional operations of the two different blocks of input data pixels. For example, the input data pixels inside the purple and pink boxes are used in common for both the first and second blocks and both the second and third blocks, respectively.

However, as can be observed, the pattern of repeated input data pixels between the first and second blocks is not the same as that between the second and third blocks, as denoted in purple and pink, respectively, in Figure 3. In other words, although some input data pixels in the two different blocks are used in common, the pattern of the repeatedly used input data pixels might vary at different blocks. In particular, although some input data pixels are commonly used for the convolutional operations of two different blocks, it is impossible to exploit the commonly used input data pixels for data reuse unless the locations of the commonly used input data pixels at each of the two consecutive blocks are fixed.

Nevertheless, if we can exploit the repeated use of input data pixels among the interblock convolutional operations, we can reduce the global buffer access for the interblock operations, as well as for the convolutions within a given block of input data pixels. In the following section, we present a novel procedure for rearranging the convolutional operations such that a group of input data pixels used repeatedly appears with a fixed regularity. By doing so, we can significantly reduce the global buffer access required for reading the input data pixels.

## 3. Proposed Method of Minimizing Global Buffer Access Using the Local Register File

### 3.1. Local Register File for Data Reuse within Each Row of the Present Input Data Block: Intrablock Register File

As shown in the preceding section, when the temporal data reuse factor is 16, the same number (16) of input data pixels should be read from the global buffer for the convolutional operations with each weight pixel, as shown in Figure 2 and Figure 3.

In the input data read pattern shown in Figure 2, notably, there is only one new input data pixel in the present set of 16 input data pixels relative to the previous set, provided that each of the two sets are to be convolved with each of the two consecutive weight pixels in a given row. In other words, once a set of 16 input data pixels has been read from the global buffer, we need to read only one more input data pixel to form the next set of 16 input data pixels unless the next set is to be applied to the weight pixel of a different row.

We claim that because 15 out of 16 input data pixels have already been read during the previous global buffer access, these 15 input data pixels do not have to be read again if we define proper registers near the 2-D MAC array. For this purpose, with the temporal data reuse factor being 16, we define 16 registers for the reuse of the 15 input data pixels. This register file, consisting of the 16 registers, is herein denoted as an intrablock register file. The term is attributable to the fact that this register file reuses the input data pixels needed within a given input data block. Specifically, the objective of defining the intrablock register file is to reuse the 15 input data pixels involved in the convolutional operations with the two consecutive weight pixels in common. With the intrablock register file, the burden of accessing the global buffer can be considerably reduced, thereby reducing the power consumption for global buffer access.

Figure 5 illustrates how the input data pixels are read from the global buffer (a) without the intrablock register file and (b) with the intrablock register file. Without the intrablock register file, 16 input data pixels should be read from the global buffer for the convolution with each weight pixel. In other words, 16 input data pixels should be read repeatedly whenever required in the convolutional operations. As shown in the red color of Figure 5b, however, the 16 input data pixels, {(0,0), (0,1), …, (0,15)}, are first read from the global buffer to be stored in the intrablock register file and to be convolved with the first weight pixel, (0,0). For the convolution with the next weight pixel, (0,1), only one input data pixel, (0,16), is read from the global buffer because the remaining 15 input data pixels, {(0,1), (0,2), …, (0,15)}, are reused from the intrablock register file. Similarly, for the convolution with the weight pixel of (0,2), only one input data pixel ((0,17)) is read from the global buffer because the remaining 15 input data pixels, {(0,2), (0,3), …, (0,16)} are reused from the intrablock register file. The intrablock register file is introduced so that we can read only one input data pixel from the global buffer instead of the entire 16 pixels for the convolutional operations with each weight pixel.

Notably, the input data pixels are always read sequentially: we first read (0,0), then (0,1), …, and finally (0,15) during the period of $t_{0}$. Similarly, the input data pixel (0,1) is read first from the global buffer during the period of $t_{1}$. Figure 5b indicates that the new input data pixel needed for the convolution with the next weight pixel should be stored in the intrablock register file at the position where the first input data pixel is stored during the previous convolution. In other words, the input data pixel (0,16) needed during the period of $t_{1}$ should be stored at the position of (0,0) used during the period of $t_{0}$. Similarly, the input data pixel (0,17) needed during the period of $t_{2}$ should be stored at the position of (0,1) used during the period of $t_{1}$.

After completing the convolutional operations with each of the weight pixels in a given row, the same procedure as described above is repeated for the weight pixels of the next row. Although we considered the case of a 3×3 matrix for the weight pixels in the previous section, the proposed method of using the intrablock register file is valid regardless of the dimensions of the weight pixel matrix. Moreover, because the convolutional operations for each row of weight pixels are independent of each other, no input data pixel is used in common for the convolutional operations with the weight pixels of different rows. This implies that the intrablock register file does not have to be defined separately for each row of the weight pixel matrix; thus, we need only one intrablock register file for the convolutional operations unless the convolutional operations are to be performed simultaneously for weight pixels in a number of rows in parallel, which is not our interest in this study.

For the 2-D MAC array, recalling that the input data pixels are spatially reused at each column of the 2-D MAC array, as explained in Section 2.1, we need to define only one intrablock register file for each column of the 2-D MAC array. The convolutional operations at the 16 MAC units in each column were performed in parallel. Each of the intrablock register file outputs should be split into 16 branches to be provided as an input to the corresponding MAC unit. In contrast, because each column of the 2-D MAC array represents one of 128 channels, we have neither input data pixels nor weight pixels being reused in each row of the 2-D MAC array. Thus, we need to define intrablock register files for which the number should be the same as the number of columns of the 2-D MAC array, that is, 128 in the example shown in Section 2.1.

Unfortunately, the data reuse based on the intrablock register file presented in this subsection (as shown in Figure 5) is not applicable because it is generalized. Therefore, the pattern of input data pixels being used in common varies depending on which block of input data pixels is being processed. This problem can be resolved if every block of the input data pixels is given in a fully rectangular shape, as shown in Figure 5, meaning that there is no input data block given in a broken rectangular shape, as shown at the center and bottom of Figure 3. In the following subsection, the convolutional operations between the input data and weight pixels are rearranged in such a way that each of all the input data blocks is given in a rectangular shape, as shown at the top of Figure 3. Then, the data reuse method based on the intrablock register file proposed in this subsection will be generalized.

### 3.2. Local Register File for Data Reuse between Two Consecutive Input Data Blocks: Interblock Register File

While the intrablock register file introduced in the preceding subsection reuses the 15 repeatedly needed input data pixels within a given input data block, this subsection introduces another register file for reusing input data pixels needed repeatedly in the convolution of different input data blocks.

Even with some input data pixels used in common in the convolution of two consecutive input data blocks, it is practically impossible to exploit the merits of reusing these repeated input data pixels because their pattern is not consistent, as shown in Figure 3. The objective of this subsection is to present a novel procedure for rearranging the convolutional operations in such a way that the repeated input data pixels appear with consistent regularity for all pairs of two consecutive input data blocks. Then, once the input data pixels have been read from the global buffer for the convolution of the first block in each pair, they never have to be read again for the second block.

Figure 6 illustrates the convolution of three input data blocks when the temporal data reuse factor is 16 with a weight dimension of 3×3. In particular, it shows how to rearrange the computational sequence of convolution in such a way that the input data pixels used in common for the convolution of two consecutive input data blocks appear with clear and simple regularity. With the original convolutional operations shown on the left-hand side of Figure 6, the pattern of the repeated input data pixels is not regular. In contrast, the sequence of computing the convolution can be properly rearranged in such a way that the input data pixels located in the second and third rows of the present input data block become identical to those in the first and second rows of the next block, respectively, as shown on the right-hand side of Figure 6.

Notably, the rearrangement of the computational sequence described above can be implemented simply by exchanging the third and fourth lines of the algorithm shown in Figure 4 with each other. This means that after the rearrangement, a group of input data pixels in a given input data block is regularly repeated in the next input data block. In other words, the group of input data pixels can be reused if they are stored in a register file as defined in this subsection. This register file will be denoted as the "interblock register file" because the data reuse in this case is applied to the convolution of different input data blocks, i.e., two consecutive input data blocks.

Because the number of input data pixels to be read from the global buffer for the convolution with each weight pixel is determined by the temporal data reuse factor, the interblock register file size is determined as follows:(1)$$\left(N_{wy}-1\right)\times \left(P+N_{wx}-1\right).$$

In the above, P denotes the temporal data reuse factor, with a weight dimension of $N_{wy}\times N_{wx}$. If the temporal data reuse factor and weight matrix are set to 16 and 3×3, respectively, the interblock register file will consist of 2×18 registers, as shown in yellow and blue in Figure 6. The second and third rows of the first block are repeated in the first and second rows of the second block at every pair of two consecutive input data blocks.

Figure 7 illustrates how the input data pixels are read from the interblock register file as well as from the global buffer for the convolution with weight pixels, assuming that the weight dimension is 3×3. Notably, only the new input data pixels are read from the global buffer because the repeated data pixels are stored in the interblock register file.

In Figure 7, the input data pixels used for the convolution at each time interval, $t_{m}^{{}^{\prime }}$ for *m* = 0, 1, 2, …, 8, are denoted by either red or blue. During each time interval, three ($N_{wx}$) sets of 16 (*P*) input data pixels are to be read either from the global buffer or the interblock register file; this takes 48 ($N_{wx}\times P$) clock cycles, as shown on the left-hand side of Figure 7. Specifically, during the time intervals of $t_{0}^{{}^{\prime }}$, $t_{3}^{{}^{\prime }}$, …, or $t_{6}^{{}^{\prime }}$, the input data pixels inside the red or blue box are convolved with the first-row weight pixels, [(0,0), (0,1), (0,2)]. In general, during the time interval of $t_{3i+j}^{{}^{\prime }}$, for *i* = 0, 1, 2, …, with *j* being either 0, 1, or 2, the input data pixels inside the colored box are convolved with the ${\left(j+1\right)}^{th}$ row weight pixels, i.e., [(*j*,0), (*j*,1), (*j*,2)].

From the above discussions, it can be observed in Figure 7 that global buffer access is needed only for the time interval of $t_{2}^{{}^{\prime }}$, $t_{5}^{{}^{\prime }}$, …, $t_{3m-1}^{{}^{\prime }}$ for *m* = 1, 2, …. In addition to these time intervals, the input data pixels are available from the interblock register file, meaning that global buffer access can be avoided by taking the input data pixels from the interblock register file instead of the global buffer. Consequently, using the interblock register file, the global buffer access can be reduced by nearly 1/3 because the first two rows in every input data block are the same as the last two rows in the previous block; as such, only the last row out of three ($N_{wy}$) rows needs to be read from the global buffer. For the very first block, because the interblock register file must be empty at the initial state, the input data pixels should be read from the global buffer for the first three consecutive intervals, $t_{0}^{{}^{\prime }}$, $t_{1}^{{}^{\prime }}$, and $t_{2}^{{}^{\prime }}$, after which global buffer access is required only once every third time interval.

Once the computational sequence of the convolution is rearranged as discussed above, the shape of every input data block becomes a whole rectangle, as shown in Figure 5. This means that the irregularity of the input data repetition pattern, a critical hindrance to the data reuse by means of the intrablock register file as introduced in the preceding subsection, is completely resolved by the rearrangement. In other words, once the computational sequence of convolution is rearranged, as presented in this subsection, data reuse can be accomplished for the two rows repeated in every pair of two consecutive input data blocks, as well as for the 15 input data pixels repeated in every row of a given input data block. This former data reuse is provided via the intrablock register file, and the latter data reuse is provided via the interblock register file.

To fully exploit the above-mentioned data reuse using the proposed intra- and interblock register files, the output feature width should be a multiple of the temporal data reuse factor, as explained in Section 2.3. Otherwise, we cannot avoid having some remaining input data pixels not included in the convolutions of the input data blocks of the entire rectangular shape. This means that some additional convolutions are required after completing the convolutions with respect to all of the regular input data blocks of an entire rectangular shape. However, the number of remaining pixels is generally far less than that included in the regular input data blocks.The exact number of output feature pixels yet to be computed for the remaining input data pixels is, as can be observed from Figure 6, at most $\left(H-\left(N_{wy}-1\right)\right)\times \left(P-1\right)$ where *H* is the height of the entire input data pixel, i.e., 6 in the case of Figure 6. In this case, to use the proposed method in the remaining input data pixels, *P* is chosen as $P=N_{rx}-\left(N_{wx}-1\right)$ where $N_{rx}$ is the width of remaining pixels. In Section 5, we discuss how the computation of the remaining pixels affects the performance of the proposed method.

Using the proposed method, global buffer access can be reduced in the blue area as shown in Figure 5 and Figure 7. In order to reuse data through the local register file, the path that originally transmitted data directly from the global buffer to the MAC array must be changed. Specifically, the MAC array should receive data from the local register file, and the local register file should be able to update data from the global buffer. However, if data is transmitted only from the local register file, the local register file cannot be used during the update process of the local register file, and a stall may occur. To solve this problem, we made it possible to transfer the data from the global buffer to the MAC array at the same time when updating the data to the local register file. In other words, data read from the global buffer can be used simultaneously in both the local register file and the MAC array. Through this, it was improved so that the operation can be performed every cycle without the stall caused by the local register file. Figure 8 shows the structure described above.

### 3.3. Integrating the Intra- and Inter-Block Register Files

This subsection presents the integration of the two proposed local register files, i.e., the intra- and inter-block register files, such that only a single hardware resource needs to be assigned to both register files.

To integrate the intra- and inter-block register files with a single hardware resource, input data pixels needed for the purpose of the intra-block register file should be provided from the inter-block register file in accordance with the required scheduling. Recalling that the inter-block register file size, (N−1)×(P+N−1), is exactly N−1 times larger than the intra-block register file size, (P+N−1), we can define N−1 intra-block register files if the contents in each row of the inter-block register file can be used for the purpose of the intra-block register file. As discussed in Section 3.2 with the explanations of Figure 7, the contents of each row of inter-block register files are exactly what is needed for the intra-block register file. The intra-block register file size, which is set to P in Section 3.1, can be extended to $P+N_{wx}-1$ to match the inter-block register file size. Specifically, the contents stored at the $\left(N_{wy}-1\right)\times \left(P+N_{wx}-1\right)$ registers of the inter-block register file are to be used as the contents of the N−1 intra-block register files, each of which is provided from the corresponding P+N−1 registers of the inter-block register file. In short, the contents of each of the N−1 rows in the inter-block register file are used as the corresponding intra-block register file.

## 4. Hardware Implementation

The objective of this section is to verify the functionality of the proposed accelerator through hardware implementation. To verify the functionality, the proposed accelerator employing the local register file that operates in accordance with the rearranged computational sequence is implemented as discussed in this section. The aim is to execute a simple example code of a deep learning neural network. We implemented the accelerator with an FPGA (XCZU9EG-2FFVB1156) mounted on an off-the-shelf evaluation board, ZCU102 (Zynq UltraScale+ MPSoC) [25], using an open hardware source, the "NVIDIA Deep Learning Accelerator" [22]. For the implementation of an accelerator for fully exploiting the merit of data reuse, the open source code was modified in such a way that the proposed local register file could be utilized in accordance with the procedure explained in Section 3.2.

Figure 9 illustrates a block diagram of the entire system, including the accelerator. The entire system consisted of an ordinary personal computer (PC) and the evaluation board, ZCU102, which comprised (1) a processing system (PS) including an ARM processor, (2) programmable logic (PL) including the proposed local register file, as well as a convolution engine, global buffer, and an interface for connecting the PS to PL, and (3) dynamic random access memory (DRAM). The ARM processor controlled the accelerator in accordance with a given deep learning application executed on an operating system (i.e., Linux in our implementation) of the ARM processor. Specifically, the ARM processor provided the instructions required for the configuration of the convolutional operations to the accelerator. From these instructions, all of the required data, i.e., both the input data and weight pixels, were transferred from the DRAM into the global buffer.

The accelerator was implemented according to the hardware description language code. In our implementation, we used Verilog software. Meanwhile, the deep learning application executed on the ARM processor could be implemented with a high-level language such as C/C++ or Python.

The instructions, which were the result of executing the application code, provided the parameter values required for the convolutional operations at each layer of the deep learning neural network in the accelerator. For instance, the dimensions of the input data and weight pixels and their addresses in the DRAM were provided as the configuration information. Then, all the data needed for the convolutional operations were transferred from the DRAM into the global buffer.

The ZCU102 board and desktop were interconnected via a serial port, whereas the Tera Term/Gimp Toolkit + Terminal (GtkTerm) was used as a terminal emulator for serial communication between the ARM processor and desktop CPU. Then, the ZCU102 board could be monitored through serial communication to the PC monitor.

Figure 10 illustrates a block diagram of the accelerator implemented on the PL of the ZCU102 board. Using the 2-D MAC array consisting of 16×128 MAC units, we exploited the spatial data reuse with a reuse factor of 16; 128-channel input data pixels were processed in parallel along each row. The proposed local register file, in which intra- and inter-block register files were integrated, was implemented near the 2-D MAC array. The size of register files was chosen as 4×20(160 Byte), which can cover both 3×3 and 5×5 weights with the reuse factor (*P*) 18 and 16, respectively, as explained in Section 3. In (Equation 1) of Section 3.2, the necessary register file size was obtained according to the weight size and the temporal data reuse factor. However, since the size of the local register file is predetermined, on the contrary, an achievable temporal data reuse factor is determined according to the weight size. Therefore, there may be differences in performance depending on the size of the weight, and the detailed results should be summarized in the result.

The register files are shared with MAC units locally by column, with the input indexed by a single IIC. Therefore, the proposed shared register files can save more resources than a per-MAC register file [17]. The accumulator shown on the right-hand side of the top part of Figure 10 summarizes all 128 multiplication results obtained from the parallel operations of the input data and weight pixels. The final result of the accumulator was stored in the DRAM. Meanwhile, to read or store any data from or to the DRAM, an advanced extensible interface bus was used.

Figure 11 shows a photograph of the entire system and corresponds to the block diagram in Figure 9. It can be observed that the PC desktop and ZCU102 boards are interconnected with a serial port cable. The monitor shows the final result of the execution of the application code, as obtained by accessing the ARM processor via the terminal emulator.

To verify the functionality of the implemented accelerator, we used an example program of the LeNet model deep learning application for recognizing Arabic numbers [26]. This example LeNet code was executed on the proposed accelerator shown in Figure 10 using the entire system architecture shown in Figure 9 and Figure 11. The application code was first compiled specifically for the implemented accelerator. The resulting executable code consisted of instructions (1) for transferring the configuration information for each layer of the LeNet deep learning network and (2) for activating the accelerator itself. Consequently, the convolutional operations required at each layer of the deep learning neural network were performed.

Figure 12 illustrates the confusion matrix obtained from the implemented accelerator. The Arabic numbers (0, 1, …, 9), shown in the vertical and horizontal axes of the confusion matrix, denote the input and output of the LeNet DNN, whereas each entry value represents the probability that the input value is inferred as the output value. Meanwhile, the pixel values for each Arabic number of the Modified National Institute of Standards and Technology database [27] were used as the input data pixels for the LeNet deep learning network.

As shown in Figure 12, the implemented accelerator employing the proposed local register file yields reliable inference probability values. Based on the results obtained from the implemented accelerator employing the proposed local register file, the functionality of the proposed accelerator is verified. In the following section, we will present exactly how many global buffer accesses can be reduced by the proposed accelerator by employing the local register file and intra- and inter-block register files.

Before delving into Section 5 to detail all the merits of the proposed accelerator employing the local register file, FPGA hardware resource utilization of the proposed accelerator is summarized here in comparison to the conventional accelerator [22]. The objective is to summarize how much extra hardware resources are required for implementing the proposed local register file on the FPGA.

Table 2 shows the FPGA hardware resource utilization when implementing the conventional [22] and proposed accelerators on the FPGA of the ZCU102 board. It has been found in our hardware implementations that the proposed accelerator for the deep learning network shown in Figure 9, Figure 10 and Figure 11 requires 81,375 units of flip-flop (FF), 72,759 units of look-up table (LUT), 100 units of block random access memory (BRAM), and 105 units of a digital signal processor (DSP) slice, while the conventional accelerator can be implemented with 74,965 units of FF, 65,640 units of LUT, 100 units of BRAM, and 100 units of a DSP slice. Consequently, the extra amount of hardware resources required by the proposed accelerator is approximately 8.6% of the FF, 10.8% of the LUT, and 5% of the DSP slice in comparison to the conventional accelerator. Specifically, the increase in the FF utilization is due to the local register file employed by the proposed accelerator for enhancing the data reuse, while the increase in both LUT and DSP slice utilization is due to the control logic for controlling the rearrangement of the computational sequence.

As discussed above, an increase in hardware resource utilization is inevitable due to the local register file and its control logic. However, the proposed accelerator fully exploits all the merits of data reuse through the proposed local register file, which is nothing but a set of FFs that consume far less power compared to the global buffer consisting of BRAMs. The advantages of the proposed accelerator are demonstrated in the following section mainly in terms of the required power consumption.

## 5. Results

### 5.1. Performance Analysis

This subsection presents the gain in global buffer access that can be obtained by the proposed method; the functionality itself was verified as discussed in the preceding section through hardware implementation. Specifically, the simulation results presented in this subsection compare the gain using the proposed and conventional methods [17,18,19,20,21,22]. In other words, we present how much more gain can be obtained in addition to the spatial and temporal data reuse provided by the conventional technique.

For simplicity, let us first assume that the output feature width is a multiple of the temporal data reuse factor, $P=20-(N_{wx}-1)$ (e.g., *P* for 3×3, 5×5, and 11×11 are 18, 16, and 10, respectively). Then, as explained in Section 3, both the intra- and inter-block register files can be applied to every input data block because all of the input data blocks are given in an entire rectangular shape. With the dimensions of the weight matrix being $N_{wy}\times N_{wx}$, the number of global buffer accesses to obtain a single block of input data pixels is $\left(20-\left(N_{wx}-1\right)\right)\times N_{wy}\times N_{wx}$ or $N_{wy}\times 20$ when the accelerator does not or does employ the proposed intra-block register file, respectively [22]. Notably, without the proposed intra-block register file, each set of $20-(N_{wx}-1)$ input data pixels must be read from the global buffer per weight pixel, whereas only one out of the $20-(N_{wx}-1)$ input data pixels would have to be read per weight pixel if the accelerator employs the proposed intra-block register file. Thus, when there is only a single input data block, the global buffer access gain provided by the intra-block register file can be written as follows:(2)$$100\times (1-\frac{20}{\left(20-\left(N_{wx}-1\right)\right)\times N_{wx}})\%$$

If the total number of input data blocks is *M*, the number of global buffer accesses to obtain the entire *M* blocks of input data pixels is $M\times \left(20-\left(N_{wx}-1\right)\right)\times N_{wy}\times N_{wx}$ or $(M+\left(N_{wy}-1\right))\times 20$ when the accelerator does not or does employ both intra- and inter-block register files, respectively. Note that the size of register files limits the inter-block data reuse. For example, if the weight size is larger than 5×5, the inter-block register files should be overwritten after the reading of the 4th row of a block, increasing global buffer access by $20\times N_{wy}\times M$. However, such large weights are seldomly used in the modern DNN structure, incurring limited harm for the proposed hardware.

Thus, when there are *M* input data blocks, the global buffer access gain provided by both the intra- and inter-block register files can be written as follows:(3)$$100\times (1-\frac{(M+\left(N_{wy}-1\right))\times 20}{M\times \left(20-\left(N_{wx}-1\right)\right)\times N_{wy}\times N_{wx}})\%$$

When the total number of input data blocks, *M*, is sufficiently larger than the number of rows in the weight matrix, $N_{wy}$, which is generally the case, then the global buffer access gain shown in (Equation 3) can be approximated as follows:(4)$$100\times (1-\frac{20}{\left(20-\left(N_{wx}-1\right)\right)\times N_{wy}\times N_{wx}})\%$$

Notably, as mentioned at the beginning of this section, the global buffer access gain shown in (Equation 2)–(Equation 5) is valid only when the output feature width is set to a multiple of the temporal data reuse factor. If this requirement is not met, the global buffer access gain decreases because of the remaining input data pixels, as discussed in the last paragraph of Section 3.2.

When the output feature width is not set to a multiple of the temporal data reuse factor, meaning that there exist some remaining input data pixels, the global buffer access gain with the intra-block register file only can be obtained as follows:(5)$$100\times (1-\frac{M\times 20\times N_{wy}+\beta }{M\times \left(20-\left(N_{wx}-1\right)\right)\times N_{wx}\times N_{wy}+\alpha })\%$$

In contrast, the global buffer access gain with both the intra- and inter-block register files can be determined as follows:(6)$$100\times (1-\frac{(M+\left(N_{wy}-1\right))\times 20+\beta }{M\times \left(20-\left(N_{wx}-1\right)\right)\times N_{wy}\times N_{wx}+\alpha })\%$$

In (Equation 5) and (Equation 6), β and α denote the number of global buffer accesses required to process the remaining input data pixels when the proposed method is used or not, respectively, and can be computed as follows:(7)$$\alpha =R_{op}N_{wy}N_{wx}$$
(8)$$\beta =R_{ip}$$

Here, $R_{i_{p}}$ and $R_{o_{p}}$ are the number of remaining input data pixels and remaining output data pixels, respectively. In addition, (Equation 2) and (Equation 3) can be obtained from (Equation 5) and (Equation 6) with α = 0 and β = 0, respectively.

As mentioned in the last paragraph of Section 3.2, the number of remaining input data pixels is, in general, far less than that of the other input data pixels in the entire M input data blocks. Thus, the impact of the remaining input data pixels on the overall global buffer access gain must decrease as the input data size increases, as shown in Figure 13.

Figure 13 illustrates the global buffer access gain provided by the proposed local register file. It can be observed that the global buffer access gain increases as either the weight dimension or input data size increases. In addition, we can observe that there are local maxima whenever the output feature width is set to a multiple of the temporal data reuse factor, that is, $20-(N_{wx}-1)$ in this study. Furthermore, as mentioned earlier, the impact of the remaining input data pixels diminishes as the input data dimension increases. The global buffer access gain becomes approximately 95%, meaning that the proposed intra- and interblock register files reduce the global buffer access down to approximately 5% for a weight dimension of 5×5.

### 5.2. Evaluation on DNN Models

This subsection presents the gain when the proposed method is applied to DNN models. The input size and weight size are different for each model or for each layer of one model. Therefore, before evaluating the model, we would like to show the gain result according to the approximate input size for the most commonly used 3 × 3 and 5 × 5 weights. At the end of this subsection, we show the results when applied to DNN models, e.g., VGG16, AlexNet, and RestNet50.

Table 3 lists the global buffer access gain and power consumption gain provided by the proposed intrablock register file. The global buffer access gain is obtained from (Equation 5), in which the remaining input data pixels and input data pixels within the regular input data blocks of the entire rectangular shape are considered. The power gain is computed under the assumption that the power consumption per global buffer access is k times larger than that per local register file access. Because the global buffer and local register file, as implemented on the FPGA shown in Section 4, are approximately 512 K and 20 K bytes, respectively, the value for k is set to 6 [12] to obtain the values for the power gain shown in Table 3. When the input data dimension is 64×64 with a weight dimension of 5×5, for instance, Table 3 shows that the power gain is approximately 61.93%, thus indicating that the power consumption for the proposed accelerator to read all the input data pixels is only approximately 40% of that of the conventional accelerator [22]. It can also be observed that the power gain and global buffer access gain increase as the data size increases.

Table 4 lists the global buffer access gain and power gain values obtained with both intra- and interblock register files. With the two local register files integrated within a single hardware resource, as explained in Section 3.3, both the global buffer access gain and power gain are significantly enhanced. If the input data size is large enough, for example, 512×512, the global buffer access gain becomes approximately 95% when the weight dimension is 5×5. Thus, the required number of global buffer accesses is reduced to approximately 1/20, whereas the power consumption for the proposed accelerator to read all the input data pixels decreases to approximately 21% of the conventional value.

Table 5 lists the performance gain of the proposed hardware evaluated on conventional DNNs such as VGG16 [28], AlexNet [1], and ResNet50 [29] models. VGG16 achieves the access gain of 86.75%, which is near the ideal gain (88.89%), thanks to its simple 3×3 convolutions. In the case of AlexNet, the access gain is slightly degraded (= 85.65%) mainly due to 11×11 with a large stride; yet the degradation is minor since most weights are either 3×3 or 5×5. On the other hand, ResNet50 achieves significantly lower access gain 45.10%, since about two-thirds of the convolution weights are 1×1 limiting pixel-wise data reuse. Still, it is shown that the proposed hardware can reduce global buffer access from 45% to 86%, resulting in significant power savings.

Lastly, we investigate the reduction in global buffer access on light-weight convolutional neural networks (CNNs) such as MobileNets [30,31] and EfficientNets [32]. These efficient CNNs consist of depth-wise separable convolution that decomposes convolution into computation along the channels and the features. The convolution computation along the channel is identical to 1×1 convolution, which does not reuse feature data via the local register file. However, the convolution computation along the feature (called depth-wise convolution) can exploit feature data reuse via the intra- and inter-block register files for the convolution window. Table 6 shows the global buffer operation gain for MobileNet-V2 [31] and EfficientNet-B0 [32]. Each bottelneck layer contains one or more depth-wise layers. In MobileNetV2, each layer contains {1,2,3,4,3,3,1} depth-wise layers, and in EfficientNet-B0, {1,2,2,3,3,4,1} are included. The table result shows the global buffer access gain for each bottleneck layer, and it can be seen that the gain is about 70% to 87%. Among them, in the case of the bottleneck layer, which has a low gain compared to other layers, about 70%, it can be seen that the reuse rate is lowered by applying stride = 2 in the first depth-wise layer among several depth-wise layers included in each layer. Note that depth-wise convolution is more data-hungry than point-wise convolution; it only utilizes one row of the 2-D MAC array to compute convolution across the weight pixels while requesting the same bandwidth for the feed of feature data. Therefore, the reduction in global buffer access is particularly essential for depth-wise convolution.

## 6. Conclusions

By introducing local register files that operate based on a novel procedure of rearranging the convolutional operations, we have presented a method for realizing a power-efficient deep learning accelerator that minimizes the global buffer access by maximizing data reuse. Compared to conventional data reuse methods [17,18,19,20,21,22], which suffer from a limited data reuse factor, the proposed technique provides a flexible and sufficiently large data reuse factor such that input data pixels do not have to be read repeatedly from the global buffer. Although the merits of the proposed technique can be fully exploited only when the output feature widths are set to multiple temporal data reuse factors, we have demonstrated that the superiority of the proposed method is guaranteed even when that condition is not met. For instance, with a weight pixel matrix of 5×5, the global buffer access gain provided by the proposed method is found to be 94.81%, 94.90%, and 94.95% for the input data dimensions of 128×128, 256×256, and 512×512, respectively, whereas the global buffer access gain in the ideal case is 96%. We have also quantified the reduction in power consumption for memory access based on the global buffer access gain provided by the proposed method. For instance, with a weight pixel matrix of 5×5, the proposed method saves approximately 79%, 79.09%, and 79.13% of the memory access power consumption for the input data dimensions of 128×128, 256×256, and 512×512, respectively, thereby indicating that the proposed accelerator consumes only approximately 1/5 of the memory access power compared to most conventional accelerators [17,18,19,20,21,22].

## Figures and Tables

**Figure 1 sensors-22-03095-f001:**
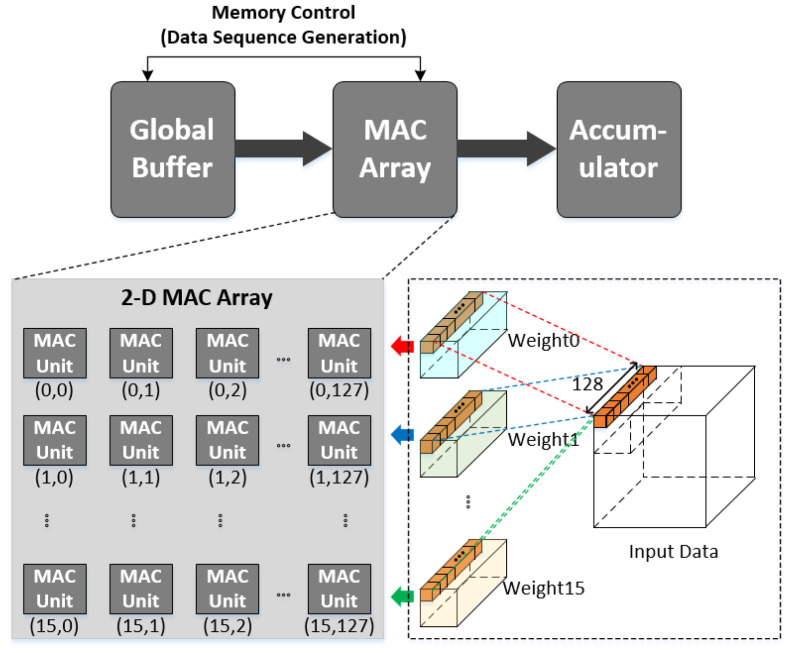
Two-dimensional multiplier-accumulator (2-D MAC) array for convolution between 128-channel input data and weights.

**Figure 2 sensors-22-03095-f002:**
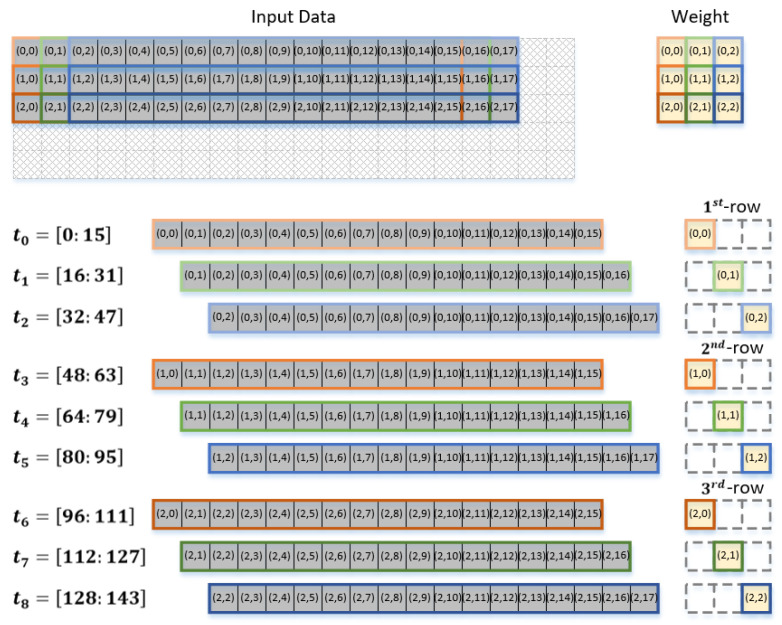
Input data read pattern for convolution with each row of 3×3 weight pixels.

**Figure 3 sensors-22-03095-f003:**
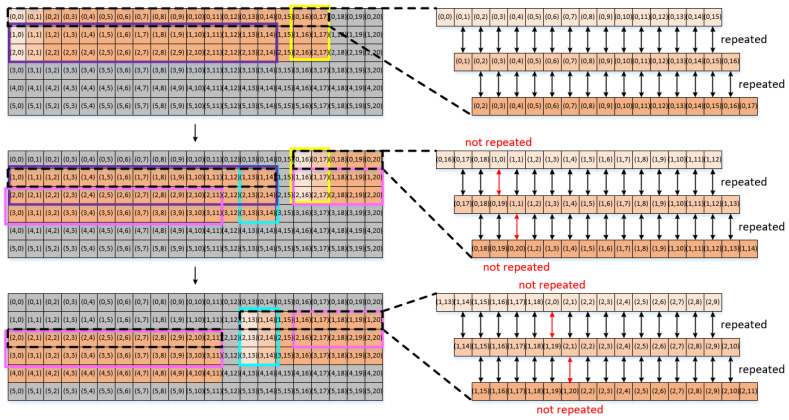
Conceptual diagram for the convolutions of the three input data blocks with the first row of 3×3 weight pixels.

**Figure 4 sensors-22-03095-f004:**
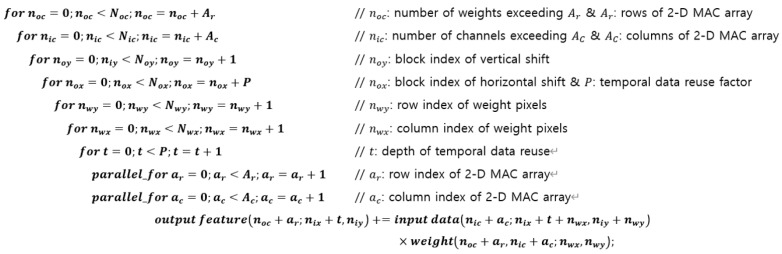
Convolutional algorithm that combines spatial and temporal data reuse.

**Figure 5 sensors-22-03095-f005:**
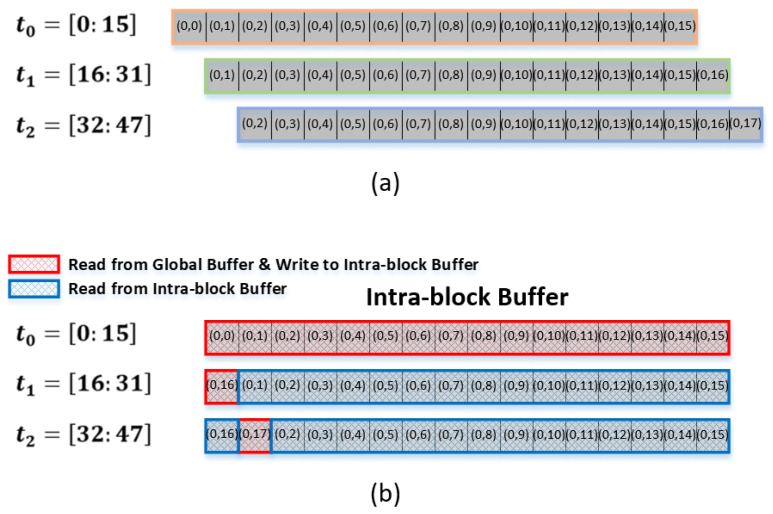
(**a**) Global buffer read pattern without Intrablock Register File: 16 Input data pixels are read whenever they are needed, (**b**) Global buffer read pattern with Intrablock Register File: Once 16 Input data pixels are read, only one new input data pixel is read afterward.

**Figure 6 sensors-22-03095-f006:**
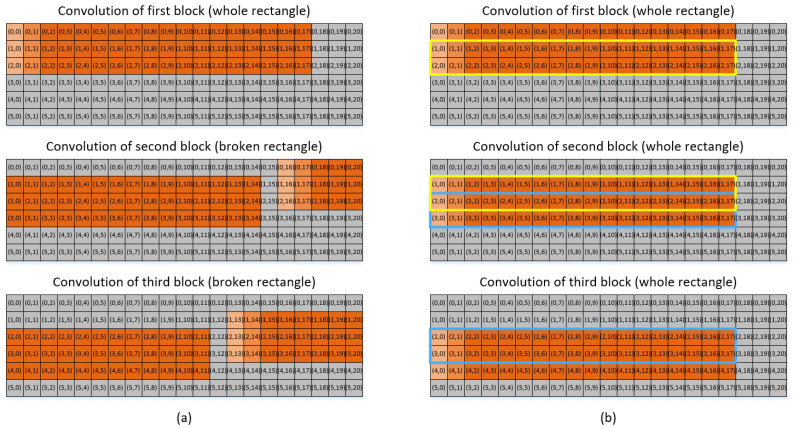
Pattern of repeatedly used input data pixels for the convolution of three input data blocks (**a**) without rearrangement of the computational sequence, (**b**) with rearrangement of the computational sequence.

**Figure 7 sensors-22-03095-f007:**
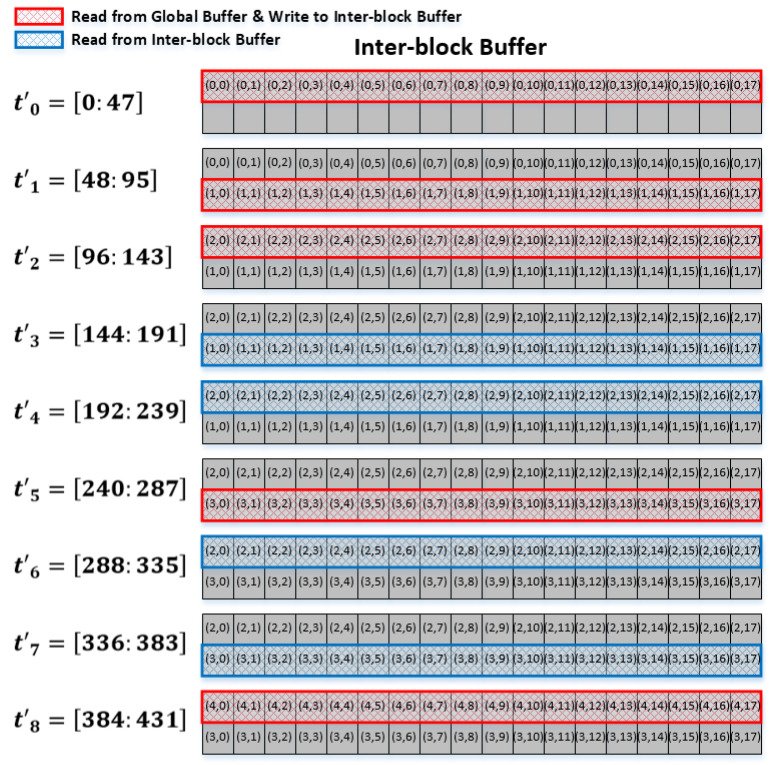
Global buffer and interblock register file access pattern.

**Figure 8 sensors-22-03095-f008:**
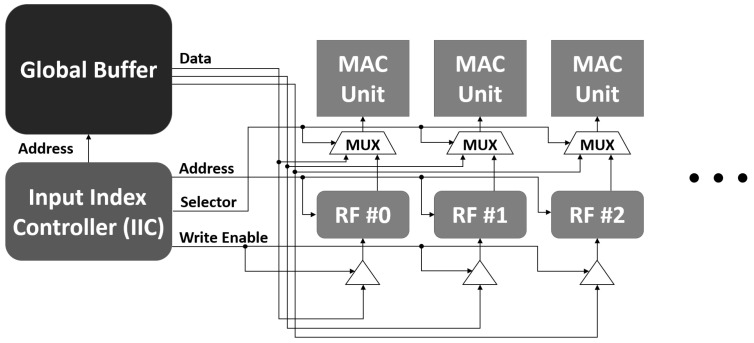
Input index controller structure to control global buffer and local register file.

**Figure 9 sensors-22-03095-f009:**
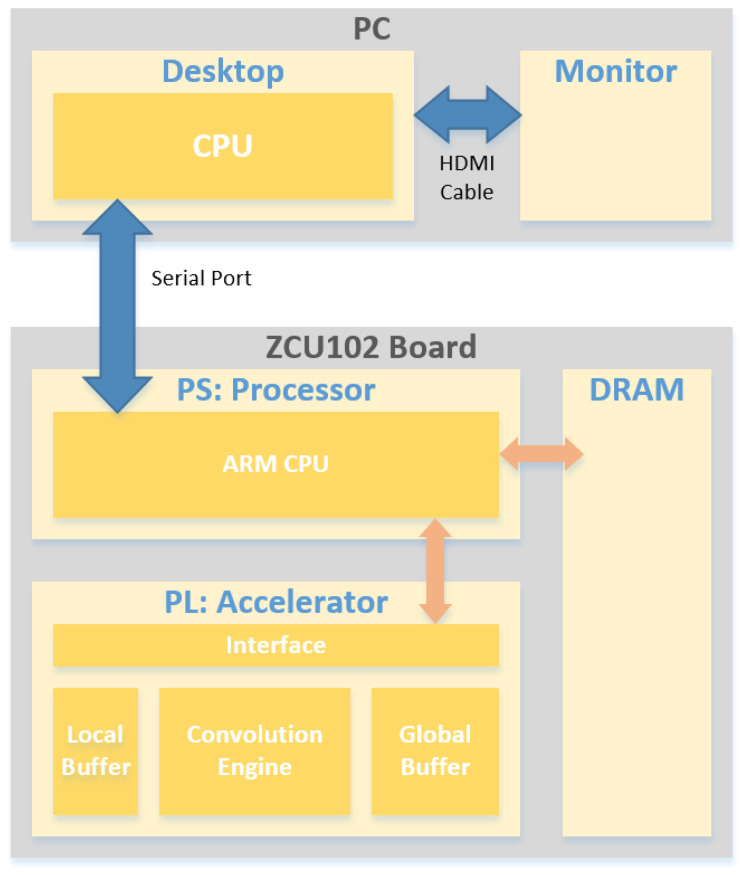
Entire system block diagram implemented with a PC and ZCU102.

**Figure 10 sensors-22-03095-f010:**
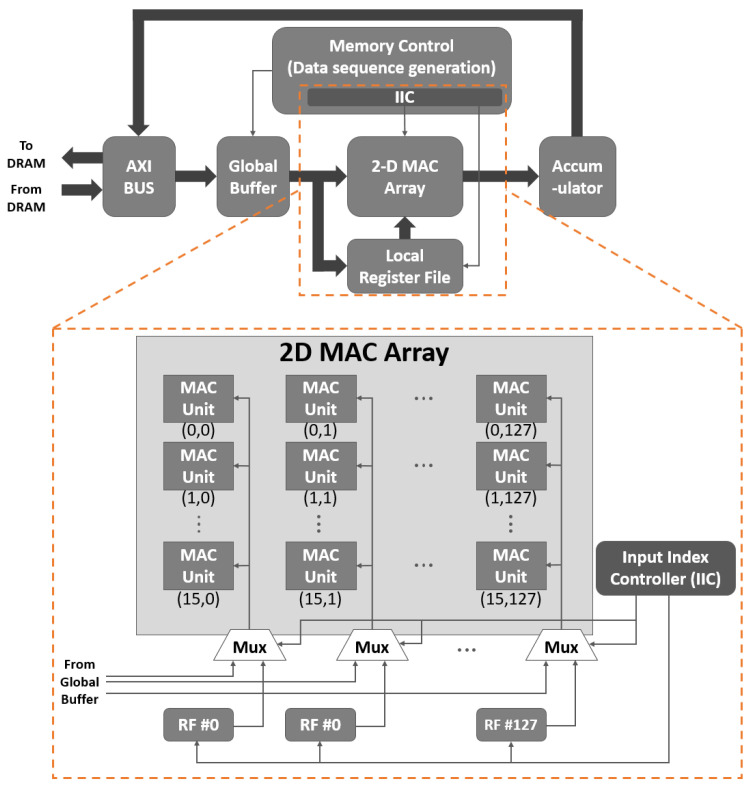
Block diagram of the proposed accelerator employing a 16×128 2-D MAC array for 16 weight sets and 128 channels.

**Figure 11 sensors-22-03095-f011:**
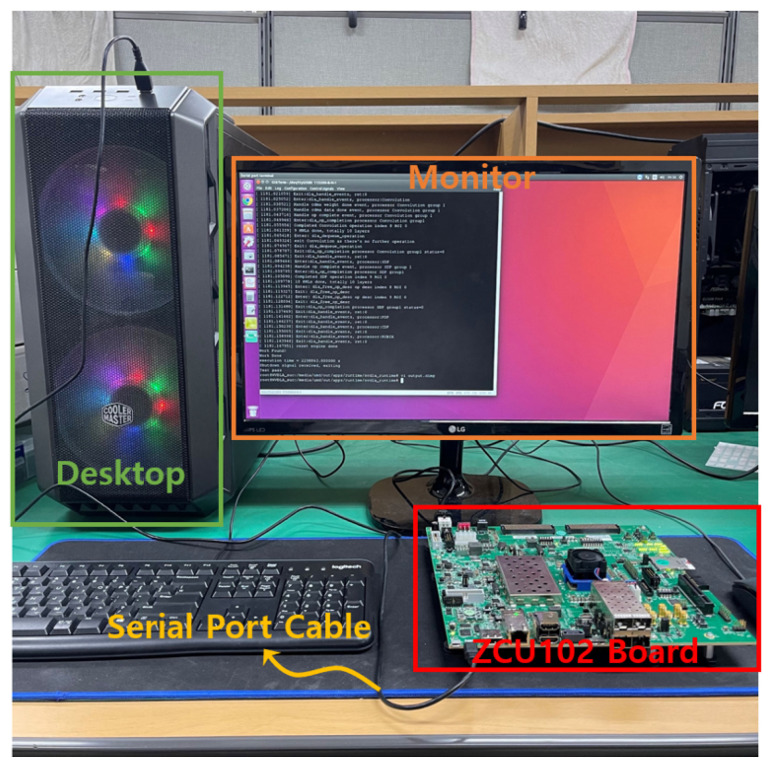
Photograph of the entire system consisting of the PC and ZCU102 board including the proposed accelerator.

**Figure 12 sensors-22-03095-f012:**
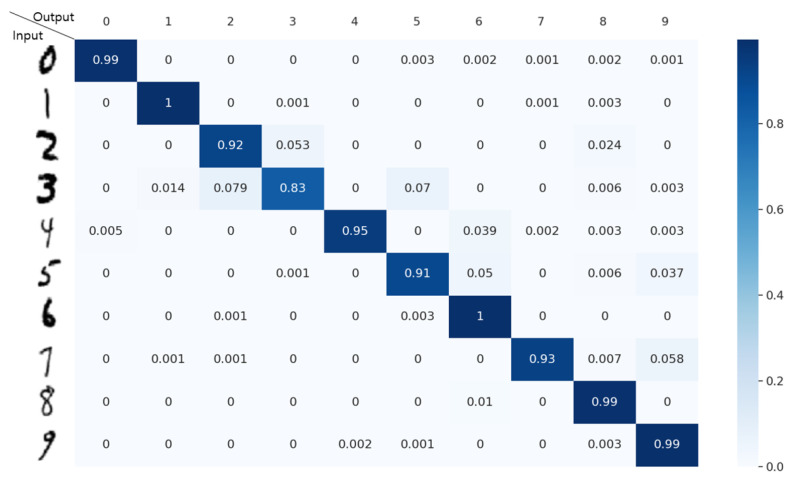
Confusion matrix of inferring Arabic numbers obtained from the implemented accelerator.

**Figure 13 sensors-22-03095-f013:**
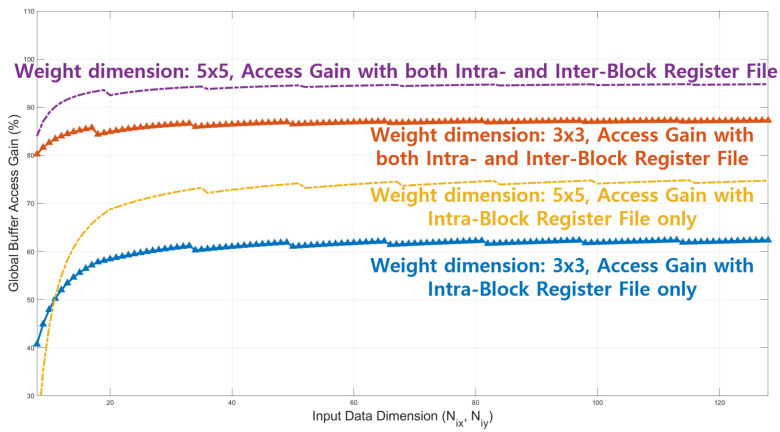
Global buffer access gain for the proposed accelerator.

**Table 1 sensors-22-03095-t001:** Index Table.

Symbols	Descriptions
$A_{r}$	# of rows of 2-D MAC array
$A_{c}$	# of columns of 2-D MAC array
*P*	Temporal data reuse factor
$N_{wx}$	Width of weight matrix
$N_{wy}$	Height of weight matrix
$N_{ix}$	Width of input data matrix
$N_{iy}$	Height of input data matrix
$N_{ox}$	Width of output feature matrix
$N_{oy}$	Height of output feature matrix

**Table 2 sensors-22-03095-t002:** Comparison of FPGA hardware resource utilization when implementing the conventional and proposed accelerator on the ZCU102 board.

	Conventional Accelerator [22]	Proposed Accelerator
FF	74,965	81,375
LUT	65,640	72,759
BRAM	100	100
DSP slice	100	105

**Table 3 sensors-22-03095-t003:** Global buffer operation gain with intrablock register file only.

	3×3 Weight		5×5 Weight	
	# of Access	Power	# of Access	Power
	Gain	Gain	Gain	Gain
8×8	40.74%	33.95%	20%	16.67%
Input Data				
16×16	56.46%	47.05%	64.44%	53.70%
Input Data				
32×32	61.04%	50.86%	72.65%	60.54%
Input Data				
64×64	62.09%	51.74%	74.31%	61.93%
Input Data				
128×128	62.37%	51.97%	74.76%	62.30%
Input Data				
256×256	62.45%	52.04%	74.90%	62.42%
Input Data				
512×512	62.48%	52.07%	74.96%	62.46%
Input Data				

**Table 4 sensors-22-03095-t004:** Global buffer operation gain with both intra- and inter-block register files.

	3×3 Weight		5×5 Weight	
	# of Access	Power	# of Access	Power
	Gain	Gain	Gain	Gain
8×8	80.25%	66.87%	84%	70%
Input Data				
16×16	85.49%	71.24%	92.89%	77.41%
Input Data				
32×32	86.57%	72.14%	94.12%	78.44%
Input Data				
64×64	87.05%	72.54%	94.60%	78.83%
Input Data				
128×128	87.28%	72.73%	94.81%	79%
Input Data				
256×256	87.39%	72.83%	94.90%	79.09%
Input Data				
512×512	87.45%	72.87%	94.95%	79.13%
Input Data				

**Table 5 sensors-22-03095-t005:** Global buffer operation gain in DNN models with both intra- and inter-block register files.

DNN	# of Access Gain	Power Gain
VGG16	86.75%	72.28%
AlexNet	85.65%	71.38%
ResNet50	45.10%	37.58%

**Table 6 sensors-22-03095-t006:** Global buffer operation gain per depth-wise layer of MobileNetV2(left) and EfficientNet-B0(right) with both intra- and inter-block register files.

MobileNetV2	# of Access Gain	Power Gain	EfficientNet-B0	# of Access Gain	Power Gain
1st DW layer	87.28%	72.73%	1st DW layer	87.28%	72.73%
in Bottleneck layer1			in MBConv1, k3 × 3		
2–3rd DW layer	70.80%	59.00%	2–3rd DW layer	74.01%	61.68%
in Bottleneck layer2			in MBConv6, k3 × 3		
4–6th DW layer	75.58%	62.98%	4–5th DW layer	70.73%	58.94%
in Bottleneck layer3			in MBConv6, k5 × 5		
7–10th DW layer	77.20%	64.33%	6–8th DW layer	86.39%	71.99%
in Bottleneck layer4			in MBConv6, k3 × 3		
11–13th DW layer	85.49%	71.24%	9–11th DW layer	75.35%	62.79%
in Bottleneck layer5			in MBConv6, k5 × 5		
14–16th DW layer	67.58%	56.32%	12–15th DW layer	74.68%	62.24%
in Bottleneck layer6			in MBConv6, k5 × 5		
17th DW layer	80.25%	66.87%	16th DW layer	81.63%	68.03%
in Bottleneck layer7			in MBConv6, k3 × 3

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
