# Peer review of "Minimizing Global Buffer Access in a Deep Learning Accelerator Using a Local Register File with a Rearranged Computational Sequence"

_sensors, 2022, doi:10.3390/s22083095_

Round 1
Reviewer 1 Report
- As Local register files are used to Minimize Global Buffer Access, so is there any effect in local register availability to local computation.
- Local register files are used to reuse the data but what if data is updated.
- When reuse data is updated then we need to update it locally and globally, so it may be a time-consuming process.
- What if local register files are not available when required?
- The size of the Global buffer and local register file may differ so by which way data is handled?
Author Response
Response to Reviewer 1 Comments
Thanks for letting us clarify the management method for Global Buffer and the local register file.
Point 1: As Local register files are used to Minimize Global Buffer Access, so is there any effect in local register availability to local computation.
>> The Input Index Controller (IIC) makes sure that the Local Register File is always available beforehand. Here is a detailed explanation of the data loading procedure. Once a bulk of input data is loaded to Global Buffer of a large capacity (512KB), input data is fetched and processed in a group of elements (called block) by 2D-MAC. In the proposed accelerator, we devise a datapath dedicated to the Local Register File so that a block of Input from Global Buffer can stage in the Local Register File. Since IIC generates a sequence of LRF accesses following the intra-/inter-block data reuse (Sec. 3.1-2), the fetched block is fully reused within the Local Register File before being kicked out. Therefore, there is no case that a new Input block requests space for LRF while a previous block is being processed. Note that the LRF size determines block size; as discussed in [Section 4], we chose LRF of 160 Byte and demonstrated practical savings in Global Buffer accesses thanks to pixel-wise data reuse.
Point 2: Local register files are used to reuse the data but what if data is updated.
>> As discussed in the previous answer, IIC makes sure that a data block is fully reused before another Input block replaces it. This extensive was possible thanks to our novel intra-/inter-block data reuse discussed on [Section 3].
Point 3: When reuse data is updated then we need to update it locally and globally, so it may be a time-consuming process.
>> We concur with the reviewer that loading Global Buffer takes a fixed amount of cycles before the computation occurs. However, there are no additional cycles for fetching an Input block to the Local Register File since the block stages in the LRF when brought from Global Buffer for the first time.
Point 4: What if local register files are not available when required?
>> As explained in answers to Points 1 and 2, IIC makes sure that the block size matches the size of the Local Register File, and the fetching takes place only when a block is fully reused.
Point 5: The size of the Global buffer and local register file may differ so by which way data is handled?
>> It is true that the size of the Local Register File (160 Byte) is much smaller than Global Buffer (512 KByte). However, a small Local Register File was enough for reducing the Global Buffer accesses since each Input pixel can be reused by the weight pixels, reducing the required memory footprint for the Local Register File. As discussed in [Sec 3.1 and 3.2], we can exploit a particular data access sequence to reduce Global Buffer access without significant resource overhead; based on our actual FPGA implementation, a modest overhead (5~11% of FPGA resources) was observed.
To improve our explanation about data management for the Global Buffer and the Local Register File, we revised the manuscript as follows:
- To strengthen a motivating example of input data reuse in Sec 2.2, we added additional comments after Fig. 2
- To provide deeper understanding about the data management between Global Buffer and the computation units, we added Fig. 8 and paragraphs explaining it.
- To explain how the proposed architecture is advanced from the baseline for data reuse within the Local Register File, we added explanation about how data stages in the LRF after Fig. 10.

Reviewer 2 Report
This paper presents an accelerator for FPGA to execute convolutional layers of Deep Neural Networks. The main contribution is a novel dataflow aiming to maximize data reuse, hence, reduce the energy due to data movement across different levels of the memory hierarchy. To achieve this goal, the authors propose an accelerator built following the NVDLA template, equipped with a local register file shared across the MAC units.
I would kindly suggest to the authors addressing the following concerns:
Point 1: It is not clear whether the proposed hardware architecture is an accelerator for any kind of deep neural networks (e.g., multi-layer perceptrons, recurrent neural networks, LSTM, etc.) and different applications like object detection and video recognition that relies also on layers different from 2D convolutions. The experimental section only considers convolutional neural networks for image classification. Clearly state the application domain of the proposed accelerator.
Point 2: More recent networks for image classification like MobileNets [1] and EfficientNets [2] address the memory bottleneck of convolutional neural networks from an algorithmic perspective, i.e., proposing a different operator called Depthwise-Separable Convolution. This operator, composed of point-wise (1x1) convolutions and depthwise convolutions, has a smaller number of parameters (hence requires less memory) but reduces the data-reuse opportunities [3]. This potentially cuts the benefits of the proposed dataflow. This limitation partially applies also to ResNets, where the reported results show halved savings due to point-wise convolutions. Which are the benefits in case standard 3x3 convolutions are replaced by depthwise convolutions? The resource overhead (10%) introduced by the proposed hardware architecture is still justified on more recent networks?
[1] 2019. Howard et al., Searching for mobilenetv3. In Proceedings of the IEEE/CVF International Conference on Computer Vision.
[2] 2019. Tan et al., Efficientnet: Rethinking model scaling for convolutional neural networks. In International conference on machine learning, PLMR.
[3] 2019. Kwon et al., Understanding Reuse, Performance, and Hardware Cost of DNN Dataflow: A Data-Centric Approach. In Proceedings of the 52nd Annual IEEE/ACM International Symposium on Microarchitecture (MICRO '52).
Author Response
Response to Reviewer 2 Comments
We thank the reviewer for improving our manuscript with expanded evaluation scope.
Point 1: This paper presents an accelerator for FPGA to execute convolutional layers of Deep Neural Networks. The main contribution is a novel dataflow aiming to maximize data reuse, hence, reduce the energy due to data movement across different levels of the memory hierarchy. To achieve this goal, the authors propose an accelerator built following the NVDLA template, equipped with a local register file shared across the MAC units. I would kindly suggest to the authors addressing the following concerns:
* It is not clear whether the proposed hardware architecture is an accelerator for any kind of deep neural networks (e.g., multi-layer perceptrons, recurrent neural networks, LSTM, etc.) and different applications like object detection and video recognition that relies also on layers different from 2D convolutions. The experimental section only considers convolutional neural networks for image classification. Clearly state the application domain of the proposed accelerator.
>> Thank the reviewers for letting us clarify the scope of our accelerator. Since the proposed accelerator is architected with a 2D-MAC array, computations based on matrix multiplication (such as Conv2D and 1x1 Conv) can be executed. In the case of vector-wise multiplication (such as depth-wise convolution), one row of the 2D MAC array is utilized to process the computation, limiting row-wise data sharing. Thanks to versatility of the 2D-MAC array architecture, the proposed accelerator can support MatMul operations of LSTM and fully-connected layers of multi-layer perceptrons along with point-wise and depth-wise convolutions of MobileNet.
Point 2: * More recent networks for image classification like MobileNets [1] and EfficientNets [2] address the memory bottleneck of convolutional neural networks from an algorithmic perspective, i.e., proposing a different operator called Depthwise-Separable Convolution. This operator, composed of point-wise (1x1) convolutions and depthwise convolutions, has a smaller number of parameters (hence requires less memory) but reduces the data-reuse opportunities [3]. This potentially cuts the benefits of the proposed dataflow. This limitation partially applies also to ResNets, where the reported results show halved savings due to point-wise convolutions. Which are the benefits in case standard 3x3 convolutions are replaced by depthwise convolutions? The resource overhead (10%) introduced by the proposed hardware architecture is still justified on more recent networks?
>> Thanks for suggesting an important evaluation direction. We concur with the reviewer that depthwise convolution of light-weight convolutional neural networks (MobileNet, EfficientNet) has become extremely popular; thus, it would be of significant interest for the readers to evaluate the proposed accelerator on depthwise convolution. To satisfy this need, we conducted an additional evaluation of the proposed hardware on the depthwise convolution of MobileNet-V2 and EfficientNet. Note that depthwise convolution lacks channel-wise data reuse due to its channel-independent computation; thus, the compute-to-data ratio becomes much lower than Conv2D operation. Only a vector unit of a 2D MAC array is utilized for computation. However, a data-reuse opportunity still exists across the weight pixels, and the proposed accelerator can take advantage of it via the Local Register File. As demonstrated in Table 6 (which is newly added), a significant amount of Global Buffer accesses are saved by the intr- and inter-block Input reuse. The resulting power gain indicates the benefits of the proposed architecture. We added in-depth discussion and explanation about this new evaluation in the revised manuscript.
